# Effects of Meteorological Conditions and Irrigation Levels during Different Growth Stages on Maize Yield in the Jing-Jin-Ji Region

**Zhixiao Zou** [1,2,3]**, Changxiu Cheng** [1,2,3,]*** and Shi Shen** [1,2,3]

[1] Key Laboratory of Environmental Change and Natural Disaster, Beijing Normal University, Beijing 100875, China
[2] State Key Laboratory of Earth Surface Processes and Resource Ecology, Beijing Normal University, Beijing 100875, China
[3] Faculty of Geographical Science, Beijing Normal University, Beijing 100875, China
[*] Correspondence: chengcx@bnu.edu.cn

**Abstract:** Maize is a major crop that is particularly sensitive to climate change. In addition, the extreme shortage of water resources threatens crop production. Thus, improving the effective utilization rate of water is an important problem to discuss. In this regard, we quantified the combined effects of meteorological conditions and irrigation levels during different growth stages on city-level maize yields in the Jing-Jin-Ji region from 1993 to 2019. The results show that the sowing period was affected by the minimum temperature, while the other growth stages were affected by the maximum temperature. At the ear stage of summer maize, when the effective irrigation rate reached the average level (52%), the inflection point of the total precipitation was 401.42 mm in the Jing-Jin-Ji region. When the total precipitation was higher than 401.42 mm, the summer maize yield decreased with the increasing total precipitation. Furthermore, the summer maize growth was significantly affected by drought at the seedling stage. At high effective irrigation rates and over long dry spells, as the mean daily temperature during dry spells increased, the maize yield easily increased. The increase in the effective irrigation rate can reverse the decrease in the summer maize yield. Moreover, the effective irrigation rate increased the maize yield with the increase rise in the temperature during longer dry spells, but the maize yield decreased with warmer temperatures during shorter dry spells. As such, our evaluation results will be useful for assessing food security and moving gradually toward achieving a water–energy–food nexus.

**Keywords:** crop; climate change; drought; water management; growth stages

## 1. Introduction

Maize is a major crop cultivated to meet the high food demand of humans and animals. It plays an important role in ensuring national food security. Agriculture is particularly sensitive to climate change [1]. However, climate change is a serious threat to crop productivity, which influences food security [2]. Meteorological factors associated with food security and food systems consist of temperature-related, precipitation-related, and integrated indicators that combine these and other variables [3,4]. The growth process of maize is vulnerable to meteorological factors, especially under some compound conditions, such as co-occurring high temperatures and low precipitation [5–8]. As an effective adaptation strategy to increase crop resilience to climate change, irrigation plays a key role in maintaining crop production and is an essential part of modern agriculture [9]. However, for urban agglomerations with scarce water resources, the irrigation level, meteorological conditions, and their interactions must be determined to help adjust the irrigation level reasonably [10]. Therefore, it is important to investigate the effects of meteorological conditions, irrigation levels, and their interactions on the maize yield.

Maize is a C4 crop that is more dependent on climatic conditions than other crops. Previous research has shown that meteorological conditions at different growth stages have different effects on the crop yield [11]. However, the research is lacking on the impact of meteorological conditions at different growth stages on the maize yield in the Jing-Jin-Ji region. For example, the sensitivity of maize to high temperatures differs depending on the growth stage, especially at the flowering and grain-filling stages [12]. High temperatures during flowering can easily lead to a decrease in the number of germinated pollen grains [13,14]. High temperatures during the grain-filling period may shorten the grain-filling time, which means that it will reduce the yield [15]. Water conditions are another important limiting factor of maize growth, and precipitation is a major factor affecting water conditions. Precipitation during pollination is well correlated with the maize yield in the northeastern United States [16].

In addition, some small-scale field trials or experiments under controlled conditions have shown that a combination of different stresses synergistically influences the crop yield [17,18]. However, in these experiments, the synergistic effects of stress combinations on yield are often overlooked [19]. Jagadish et al. [20] found that combined water and heat stress have a greater effect on critical physiological processes than individual stress. Combined effects cannot be explained or directly inferred from plant responses to individual stress [21,22]. Meanwhile, crop modeling is also widely used in crop yield research. However, owing to the large number of uncertain parameters and the lack of crop yield data from agricultural stations, it is difficult to calibrate the crop model. On the contrary, statistical models can solve these problems. The main advantages of statistical models are not limited by field calibration data and their transparent assessment of model uncertainties [3]. Therefore, it is necessary to use a statistical model to study the effect of the combined stresses from meteorological conditions at different growth stages on the maize yield.

Although crops are exposed to meteorological conditions at the same growth stage, the yield may be affected differently at different irrigation levels [23,24]. Irrigation can directly relieve crop water stress by partially alleviating the negative impact of adverse meteorological conditions on the crop yield [25–29]. However, it can reduce heat stress via surface cooling [30]. However, the need for irrigation varies between the growth stages. A few studies have investigated the impact of the interaction between meteorological conditions and effective irrigation on the summer maize yield.

The Jing-Jin-Ji region, one of the three large regional economic communities in China, is located in semihumid and semiarid areas. Owing to the limited water resources and increasing water demand, there is a sharp conflict between urban development and limited water resources [31]. The Jing-Jin-Ji region has always been deficient in water resources, and groundwater accounts for more than 70% of the regional water supply. Over the past 50 years, owing to natural conditions and the extensive groundwater overexploitation, the exploitation rate of deep groundwater has decreased by 0.99 m per year. Consequently, the deepest groundwater reached 107.38 m in the Jing-Jin-Ji region. Moreover, agricultural irrigation is in great demand. Generally, it consumes more than 50% of the total water consumption [32]. Specifically, from 2000 to 2016, agricultural water consumption accounted for 32.9% of the total water consumption in Beijing Municipality, 53.1% of the total water consumption in Tianjin Municipality, and over 70% of the total water consumption in Hebei Province. Water resources move from the agricultural sector to the industrial sector [33,34], which puts enormous pressure on food security. Therefore, for the Jing-Jin-Ji region, it is necessary to study effective irrigation for food security.

In this study, we applied a mixed-effects statistical model to examine the interactive effects of the temperature and precipitation and the role of irrigation on crop yields during different growth stages from 1993 to 2019 in the Jing-Jin-Ji region. We considered two sets of meteorological indexes to capture different physiological mechanisms: (1) meteorological conditions during the different growth stages, such as the mean temperature $T_i$ ($i$ represents the different growth stages) and total precipitation $P_i$, and (2) short-term extreme

meteorological conditions (i.e., drought and heat) during the different growth stages, such as consecutive dry days $CDD_i$ and the mean daily temperature during consecutive dry days $T_{CDD,i}$. The specific objectives of this study were (1) to determine the effects of two meteorological indexes on the summer maize yield at different growth stages, (2) to investigate the influence of interactions between the two meteorological indexes and effective irrigation on the summer maize yield, and (3) to identify the influencing mechanism and provide policy recommendations for better adaptation to maintain and even increase the maize yield.

Section 1 introduces the study area, defines the data sources, and explains the research methods. Section 2 presents the results. Finally, Section 3 presents the conclusions and discussion.

## 2. Materials and Methods

### 2.1. Study Area

The Jing-Jin-Ji region is composed of the provincial-level administrative units in China, which includes Beijing, Tianjin, and the Hebei Province. It is located in the central part of the North China Plain, between 113°27′–119°50′ E and 36°05′–42°40′ N, and is one of the most important agricultural production areas in China (Figure 1). It is the dominant area for agricultural production and groundwater depletion, accounting for 44% of the total area and approximately 90% of the total grain production in the Jing-Jin-Ji region. Summer maize is the main food crop in the region. It has a temperate humid semiarid continental monsoon climate with an annual total precipitation of 365.6 mm and an annual mean temperature of 25.1 °C during the summer maize growing season (during 1993–2019). Generally, summer maize is sown in early June and harvested in mid-September.

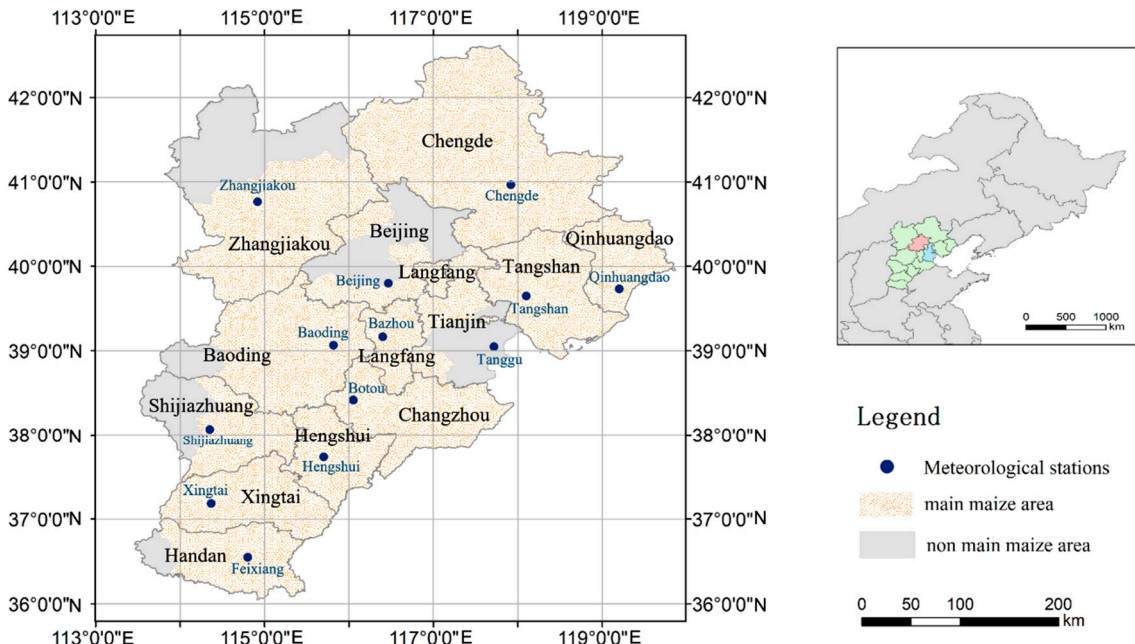

**Figure 1.** General situation of maize production in the Jing-Jin-Ji region.

### 2.2. Data

In this study, two types of datasets were used: meteorological and agricultural data. Meteorological data were collected by the National Meteorological Information Centre of the China Meteorological Administration. Agricultural data were collected using the *China Statistical Yearbook*. Detailed descriptions of the data screening are presented as follows.

Historical observed meteorological data consisting of the maximum temperature, minimum temperature, mean temperature, and precipitation on a daily timescale were obtained for the period 1993–2019 at 13 stations. The growing season of summer maize

was divided into four stages [35]: the sowing stage (6.05–6.19, abbreviated *sow*), seedling stage (6.20–7.10, abbreviated *seed*), ear stage (7.11–8.10, abbreviated *ear*), and flowering to maturity stage (8.11–9.16, abbreviated *ftm*). Some meteorological factors were chosen to reflect air temperature and soil water availability during the different growing stages: mean (abbreviated *mean*), maximum (abbreviated *max*), and minimum daily air temperature (abbreviated *min*), and average and total precipitation over the different growing stages. In addition, we chose two climatic indexes to represent short-term water and heat stresses during the growing season. One is the maximum number of consecutive dry days with daily precipitation of less than 2 mm (*CDD*) during the different growing stages, which reflects the length of the longest dry spell at each stage and measures extreme precipitation and seasonal droughts [36,37]. The other was the mean daily temperature during *CDD* ($T_{CDD}$). *CDD* and $T_{CDD}$ contain conditions for potential simultaneous damaging heat and water stress.

The agricultural data included maize yield, effective irrigated area, and total crop area from the municipal statistical yearbook. In the Jing-Jin-Ji region, winter wheat and summer maize were planted in cycles. In this study, we chose the summer maize yields (kg/ha) from 1993 to 2019. The irrigation level of the farmland was expressed by the effective irrigation rate. To accurately measure the overall characteristics of farmland irrigation, the effective irrigation rate (*irrate*) was calculated using the ratio of the effective irrigation area to the total sown area of crops.

*2.3. Methods*

How maize yields varied with the selected meteorological factors, climatic indexes, and management (i.e., effective irrigation rate) was examined explicitly using mixed-effects statistical models, including the interactions of these drivers. The mixed-effects models account for fixed and random effects. They also include group information. Fixed effects mean that the impacts of these variables on maize yields are constant for one city. By contrast, random effects mean that the impacts are variable, which extends the reliability of the inferences. In addition, this model was confirmed to be a good trade-off for use at the city scale.

For each growing stage, the meteorological factors and extreme meteorological indexes were analyzed separately in two mixed-effects statistical models. None of the variables in the models were detrended. Instead, an independent variable *t* was added to fit the trends resulting from the climate change and technological advances. The continuous variable *t* is the years elapsed from 1993. The fixed factors also included the precipitation-related indicator $P_i$ and the temperature-related indicator $T_{i,k}$ of the different growing stages for meteorological conditions, and $CDD_i$ and $T_{CDD,i}$ of the different growing stages for extreme meteorological conditions. The irrigation-related indicator *irrate* was also included as an independent variable. The two-way interactions among the factors' temperature, precipitation, and irrigation were added as fixed parts of the model. The two mixed-effects statistical models read as

$$Y_{jt} = \alpha_0 + \alpha_1 P_{ijt} P_{ijt} + \alpha_2 P_{ijt} + \alpha_3 T_{i,kjt} + \alpha_4 P_{ijt} irrate_{jt} + \alpha_5 irrate_{jt} + t + b_t + b_j + e_{jt} \quad (1)$$

$$Y_{jt} = \beta_0 + \beta_1 CDD_{ijt} + \beta_2 TCDD_{ijt} + \beta_3 CDD_{ijt} * TCDD_{ijt} + \beta_4 CDD_{ijt} irrate_{jt} + t + b_t + b_j + e_{jt} \quad (2)$$

where *t* is the year elapsed from 1993, $Y_{jt}$ is the maize yield in city *j* in year *t*, $P_i$ is a precipitation-related indicator (*i* = 0 for sowing, *i* = 1 for seedling, *i* = 2 for ear, and *i* = 3 for flowering to mature), and $T_{i,k}$ is a temperature-related indicator (*i* = 0 for sowing, *i* = 1 for seedling, *i* = 2 for ear, *i* = 3 for flowering to maturity, *k* = 0 for minimum temperature, *k* = 1 for max temperature, and *k* = 2 for mean temperature). Furthermore, $\alpha_i$ and $\beta_i$ explain the effects of $P_i$, $T_{i,k}$, $irrate_{jt}$, and $CDD_i$, $TCDD_i$, $irrate_{jt}$, and their interactions on maize yield under different meteorological conditions. In these models, we added the random effect $b_t$, $b_j$, $e_{jt}$ among the years and cities. $b_t$ and $b_j$ account for the random effect relevant to the time- and city-level groups, respectively, which means that they have different effects

for each year and city. For example, $b_j$ can account for the spatial correlations between meteorological factors and irrigation levels, which are spatially varying factors. $e_{jt}$ is the measurement or process error. We were not interested in the random part, but $b_t$, $b_j$ explains how the random samples were collected. If not included in the model, the covariation from the random factors produces pseudoreplication in the estimates of fixed factors.

## 3. Results

On the basis of the two mixed-effects statistical models, we obtained the results presented in Table 1.

**Table 1.** Model predictors, estimates, and significance (*p*) from the two linear mixed-effects models for maize under (a) meteorological conditions (i.e., $T_{i,k}$, $k = 0$ for minimum, $k = 1$ for maximum, and $k = 2$ for the mean) and (b) extreme meteorological conditions during the different growth stages (i.e., $i = sow$, $seed$, $ear$, and $ftm$), where *irrate* is the effective irrigation rate and $t$ is the years elapsed from 1993. The conditional $R^2$ provides information on the goodness of fit of the entire model to data, including fixed and random factors.

| Predictor | Symbol in Equation (1) | Sow | | Seed | | Ear | | ftm | |
|---|---|---|---|---|---|---|---|---|---|
| | | Estimate | *p* | Estimate | *p* | Estimate | *p* | Estimate | *p* |
| | | (a) Meteorological conditions results | | | | | | | |
| Intercept | $\alpha_0$ | $2.726 \times 10^3$ | 0.0034 | $6.651 \times 10^3$ | <0.001 | $1.174 \times 10^4$ | <0.001 | $9.513 \times 10^3$ | <0.001 |
| $P_i \times P_i$ | $\alpha_1$ | $2.881 \times 10^{-2}$ | 0.4771 | $-3.756 \times 10^{-3}$ | 0.4727 | $-3.288 \times 10^{-3}$ | 0.0421 | $-3.553 \times 10^{-3}$ | 0.5522 |
| $P_i$ | $\alpha_2$ | $-4.153$ | 0.5593 | 9.917 | 0.0095 | 5.158 | 0.0231 | $8.044 \times 10$ | 0.0598 |
| $T_{i,k}$ | $\alpha_3$ | $8.248 \times 10$ ($k = 0$) | 0.0498 | $-9.468 \times 10$ ($k = 1$) | 0.0123 | $-2.704 \times 10^2$ ($k = 1$) | <0.001 | $-2.157 \times 10^2$ ($k = 1$) | 0.0001 |
| $P_i \times irrate$ | $\alpha_4$ | 3.518 | 0.7453 | $-1.260 \times 10$ | 0.0392 | $-7.381$ | 0.0477 | $-1.348 \times 10$ | 0.0703 |
| *irrate* | $\alpha_5$ | $1.357 \times 10^3$ | 0.0407 | $2.054 \times 10^3$ | 0.0024 | $2.872 \times 103$ | 0.0012 | $3.167 \times 10^3$ | 0.0022 |
| Number of observations | | 351 | | 351 | | 351 | | 351 | |
| Conditional $R^2$ | | 0.662 | | 0.716 | | 0.741 | | 0.696 | |

| Predictor | Symbol in Equation (2) | sow | | seed | | ear | | ftm | |
|---|---|---|---|---|---|---|---|---|---|
| | | Estimate | *p* | Estimate | *p* | Estimate | *p* | Estimate | *p* |
| | | (b) Extreme meteorological conditions results | | | | | | | |
| Intercept | $\beta_0$ | - | - | $7.893 \times 10^3$ | <0.001 | - | - | - | - |
| CCD | $\beta_1$ | - | - | $-3.799 \times 10^2$ | 0.0013 | - | - | - | - |
| $T_{\mathrm{CDD},i}$ | $\beta_2$ | - | - | $-1.007 \times 10^2$ | 0.0288 | - | - | - | - |
| CCD $\times T_{\mathrm{CDD},i}$ | $\beta_3$ | - | - | 9.905 | 0.0234 | - | - | - | - |
| CCD $\times$ *irrate* | $\beta_4$ | - | - | $1.678 \times 10^2$ | 0.0003 | - | - | - | - |
| Number of observations | | | | 351 | | | | | |
| Conditional $R^2$ | | | | 0.674 | | | | | |

### 3.1. Maize Yield Responses to Meteorological Conditions and Irrigation Levels during the Different Growth Stages

The temperature-related indicator $T_{i,k}$ of the growing stages affected the maize yield differently (Table 1a). During the sowing stage, it was mainly affected by the minimum temperature, and an increase in the minimum temperature can increase the yield per unit of the maize. The total of 82.48 kg/ha maize increased with every 1 °C increase in the temperature (Table 1a). At the seedling, ear, and flowering stages, it was mainly affected by the maximum temperature, and the increase in the maximum temperature reduced the maize yield. For instance, under other conditions, the loss of maize per 1 °C increase in the maximum temperature during the seedling stage was 94.68 kg/ha (Table 1a).

The precipitation at the seedling and ear stages had a significant effect on the maize yield. Especially at the ear stage, the precipitation had different effects on the maize yield under different irrigation levels (Figure 2). If the effective irrigation rate reaches the average level (= 52%), the inflection point of the precipitation is at $P_{earjt} = 401.42$ mm. That is, when the precipitation at the ear stage was lower than 401.42 mm, the summer maize yield increased with the increase in the precipitation, whereas when the precipitation was higher than 401.42 mm, the summer maize yield decreased with the increase in the precipitation.

Furthermore, when the effective irrigation rate reached the normal level ($49\% \leq irrate_{jt} \leq 58\%$), the inflection point of the precipitation ranged from 266.73 mm ($irrate_{jt} = 58\%$) to 468.76 mm ($irrate_{jt} = 49\%$). When the effective irrigation rate was higher, the inflection point of the precipitation was lower. This means that higher effective irrigation rates are beneficial in preventing droughts.

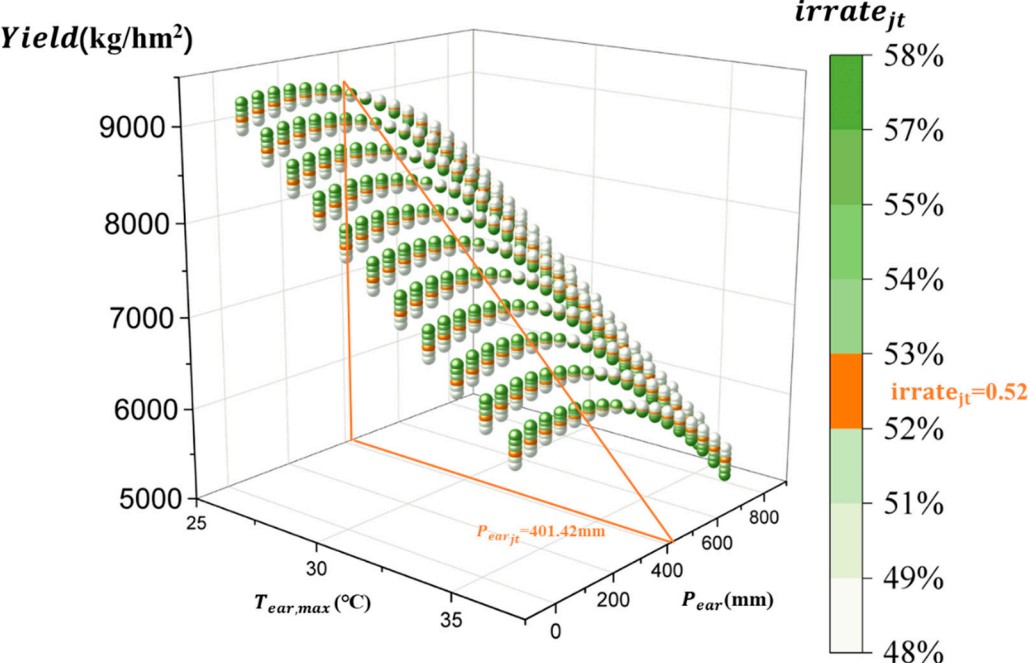

**Figure 2.** Response of crop yield *Y* to the maximum daily temperature and total precipitation during the ear stage (the orange sphere represents $irrate_{jt} = 52\%$).

### 3.2. Maize Yield Responses to Extreme Meteorological Conditions and Irrigation Levels during Different Growth Stages

The summer maize growth was significantly affected by extreme meteorological conditions at the seedling stage. At the average irrigation level, lengthening the dry spell by 1 day reduced the yields at both low and high temperatures (Figure 3a). For example, when the effective irrigation rate reached the average level ($irrate_{jt} = 52\%$) at a mean temperature of 20 °C, the loss of maize per day increase in temperature was 94.63 kg/ha (Table 1b). If the effective irrigation rate was enhanced, the maize yield could be increased (Figure 3a).

However, the maize yield responded differently to extreme meteorological conditions under different levels of effective irrigation rates (Figure 3b–f). At high effective irrigation rates and long dry spells, as $T_{CDD}$ increased, the maize yield easily increased. Figure 3b shows that at $irrate_{jt} = 80\%$, with the increase in CDD, the increase in the effective irrigation rate dampened and sometimes reversed the decrease in the summer maize yield. This means that irrigation reversed the negative impacts of drought on maize. By contrast, at low effective irrigation rates and long dry spells, with the increase in $T_{CDD}$, the maize yield did not improve as much as it did with the increase in the high irrigation rate. Furthermore, the effective irrigation rate increased the maize yield with an increase in the temperature during longer but not shorter dry spells (Figure 3a).

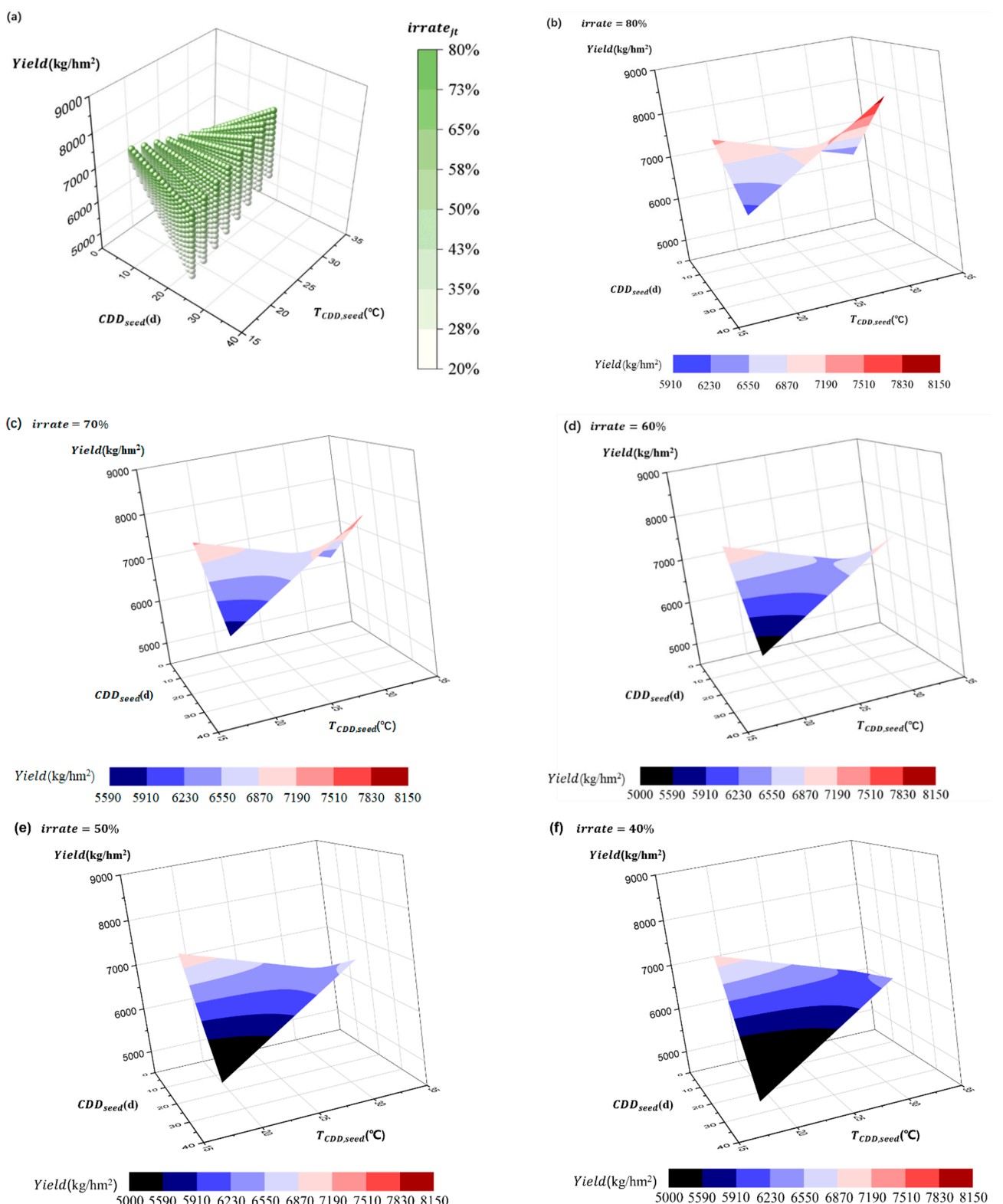

**Figure 3.** The response of maize yield to $CDD$ and $T_{CDD}$ (**a**) during the seedling stage, (**b**) at $irrate_{jt} = 80\%$ (with a saddle-shaped spatial distribution of yield), (**c**) at $irrate_{jt} = 70\%$, (**d**) at $irrate_{jt} = 60\%$, (**e**) at $irrate_{jt} = 50\%$, and (**f**) at $irrate_{jt} = 40\%$.

## 4. Discussion

Maize growth appears to be most closely related to meteorological factors, such as the temperature and precipitation [38]. The results of this study show that if the effective

irrigation rate reaches the average level ($irrate_{jt}$ = 52%), the inflection point of the precipitation is at $P_{ear\,jt}$ = 401.42 mm. That is, when the precipitation at the ear stage is lower than 401.42 mm, the summer maize yield increases with the increasing precipitation, whereas when the precipitation is higher than 401.42 mm, the summer maize yield decreases with the increasing precipitation. This may be related to the soil moisture content. Precipitation directly affects soil moisture content. Previous studies [39,40] have shown that when soil moisture contents are lower or higher than certain thresholds, crop aboveground biomass and grain yields may be reduced. Wang et al. [41] indicated through crop model simulations that the optimal yield of maize growth and actual observations usually occur when the total precipitation in the summer maize growing season is between 300 and 500 mm in clay soil in the Huang-Huai-Hai Plain. Similarly, Xu et al. [42] estimated a mean growing season optimal precipitation threshold of 503 mm in the United States. Therefore, the growth of summer maize is affected by both low and high precipitations.

Compared with the other growth stages, the seedling stage of the summer maize was significantly affected by drought in the Jing-Jin-Ji region. This was in agreement with the previous results of field experiments at an agricultural meteorological experimental station in Liaoning Province [43]. This is due to the fact that at the seedling stage, the growth and development of maize is most vigorous and the water demand is the largest. In addition, maize reproductive organs develop in the tender stage, and their resistance to external adverse environmental conditions is poor.

The results of the two analyses based on meteorological or extreme meteorological conditions provide complementary information. The extreme meteorological model had high explanatory power, just slightly lower than the meteorological model (67.4% vs. 71.6% variation explained for maize at the seedling stage; Table 1), indicating that unfavorable conditions over short durations are a key point to maize yield. Soil moisture plays a more important role at the start of the longest dry period during the seedling stage than at the start of the seedling stage, which partly explains the slightly lower yield performance.

Although the climate can influence the soil nutrient content and water-holding capacity, the soil types in our study area were mainly clay and loam soil, which have a high soil fertility, nutrient-preserved capability (e.g., N and K), and water-retaining capacity compared with sandy soil [44–46]. Therefore, we did not consider the effects of the soil type, which is another limitation of this study. Furthermore, the relationships between meteorological factors and irrigation levels might vary with the spatial variations in our study area. Despite that this part was included in the model used in this study, it was not the focus of this study. Thus, we can study it as a research direction in the future.

Irrigation can not only reverse the negative impacts of drought on maize in some conditions but also moderate the effect of temperature increases on the maize yield, especially over longer dry spells (Figure 3b). This finding is in agreement with those of previous similar studies in the United States [9,24,28]. This might contribute to the benefit of irrigation cooling. Water application can decrease the soil temperature and create a cooling effect on the soil by increasing the soil moisture [47,48]. In the Jing-Jin-Ji region, most cities belong to areas with high effective irrigation rates. Moreover, we found that increasing the effective irrigation rate effectively alleviated the hard constraints of drought (Figure 3). Therefore, the government should increase investments in the construction of farmland water conservancy facilities and comprehensively improve the effective irrigation rate, which will also enhance the quality and efficiency of the wheat production.

## 5. Conclusions

Precipitation and effective irrigation rates were the main factors that affected the summer maize yield in the Jing-Jin-Ji region. At the ear stage of the summer maize, if the effective irrigation rate reached the average irrigation level ($irrate_{jt}$ = 52%), the inflection point of the precipitation was at $P_{ear\,jt}$ = 401.42 mm, which follows an inverted-U curve. When the precipitation was higher than 401.42 mm, the summer maize yield decreased with the increasing precipitation.

The growth of the summer maize was significantly affected by drought at the seedling stage. At the average irrigation level, whether at high or low temperatures, a lengthening of the dry spell by 1 day reduced the yields. The maize yield can be increased if the effective irrigation rate is increased. However, under high effective irrigation rates and prolonged drought, the maize yield will increase more readily with an increasing $T_{CDD}$. By contrast, under low effective irrigation rates and prolonged drought, the increase in the maize yield will not be as high as with an increase in the high irrigation rates. In addition, the effective irrigation rate will increase the maize yield with an increase in the temperature during longer dry spells.

The effects of the temperature on the summer maize yield in the Jing-Jin-Ji region differed between the growth stages. First, the sowing period is mainly affected by the minimum temperature, and an increase in the minimum temperature can increase the maize yield. Second, the seedling, ear, and flowering stages are mainly affected by the maximum temperature, and an increase in the maximum temperature will reduce the maize yield.

**Author Contributions:** Conceptualization, Z.Z. and C.C.; methodology, Z.Z.; software, Z.Z.; validation, Z.Z.; formal analysis, Z.Z.; investigation, Z.Z.; resources, Z.Z.; data curation, Z.Z.; writing—original draft preparation, Z.Z.; writing—review and editing, S.S. and C.C.; visualization, Z.Z. and C.C.; supervision, S.S.; project administration, C.C.; funding acquisition, C.C. All authors have read and agreed to the published version of the manuscript.

**Funding:** This research was funded by the National Key Research and Development Plan of China (grant No. 2019YFA0606901) and the National Natural Science Foundation of China (grant No. 42041007).

**Institutional Review Board Statement:** Not applicable.

**Informed Consent Statement:** Not applicable.

**Data Availability Statement:** Data can be provided on request.

**Conflicts of Interest:** The authors declare no conflict of interest.

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
