# Peer review of "Effects of Meteorological Conditions and Irrigation Levels during Different Growth Stages on Maize Yield in the Jing-Jin-Ji Region"

_sustainability, doi:10.3390/su15043485_

Round 1

Reviewer 1 Report

This study used mixed models to investigate how meteorological conditions and irrigation levels at different growing stages influence maize yields in the Jing-Jin-Ji region, in China. The topic is interesting and important. But the manuscript needs to be proofread by an English native speaker. There is much-repeated information in the results, discussion and conclusion sections. A deep and thoughtful discussion is required in which the results are deeply interpreted. Thus, I recommend a major revision before further consideration. Moreover, the paper needs to be carefully revised according to the following detailed suggestions.

Line 24: “during longer but not shorter dry spells.” This sentence is confusing.

Introduction: The background is developed based on limited references which are mostly from China. Have the models been used to monitor the effects of climate or water levels on crop yield in other countries? What methods have been used?

Figure 1: Coordinates are required in this figure.

Line 118: Three or two?

Line 250: “300-500 cm” This value is determined by the soil type, nutrients and health. thus, it is necessary to indicate the soil types in the studied region. In this study, the relationships between climate variables and irrigation levels might vary with the spatial variations in the region. Did you consider the spatial correlations in the mixed model as residuals? This can be a future research direction. The following references which consider the effects of climate and different agriculture management strategies on crop growth, water use, soil nutrients and water holding capacity would be useful for you to discuss the results.

Zhao, D., Eyre, J. X., Wilkus, E., de Voil, P., Broad, I., & Rodriguez, D. (2022). 3D characterization of crop water use and the rooting system in field agronomic research. Computers and Electronics in Agriculture202, 107409.

Zhao, D., Wang, J., Zhao, X., & Triantafilis, J. (2022). Clay content mapping and uncertainty estimation using weighted model averaging. Catena209, 105791.

Zhao, D., Arshad, M., Li, N., & Triantafilis, J. (2021). Predicting soil physical and chemical properties using vis-NIR in Australian cotton areas. Catena196, 104938.

Zhao, D., Arshad, M., Wang, J., & Triantafilis, J. (2021). Soil exchangeable cations estimation using Vis-NIR spectroscopy in different depths: Effects of multiple calibration models and spiking. Computers and Electronics in Agriculture182, 105990.

Line 280-286: The results, discussion and conclusion sections have a lot of repeat information. In the discussion, instead of simply repeating the results, a deep and thoughtful discussion is required. This study has many great findings but has not been compared with the literature. And what are the limitations of this study and the future directions? For the conclusion, it might be better to condense the content.

Reviewer 2 Report

I revised the ms. "Effects of meteorological conditions and irrigation levels during different growth stages on maize yield in the Jing-Jin-Ji region" an in my opinion is a good research, well done. But, I really believe that the discussion could be more explored. I listed below some minor suggestions.

Key-words: please, apply differents words from the title

Introduction:

Line 35-36: check the correct citation form

Figures 2 and 3: please imprive the number size at irrate and yield scale. It is difficult to read.

Discussion

I suggest the authors to improve the discussion, including more studies even with others cultures/countries.

In general, the paper is well done, but the discussion has to be improved. The authors has a rich data that has to be more explored.

Reviewer 3 Report

The article is interesting since it discuss scientific literature indicating some limitations of the traditional meteorological conditions and irrigation levels. The article is clear in the display of backgrounds as well as in results. Moreover, it also shows a relevant literature research even if the only 11 on 66 cited articles have been published in the last 5 years. But Discussion very poor and must be improved. The article is suitable for publication after minor revision.

Round 2

Reviewer 1 Report

Authors have well addressed all my comments.

Reviewer 2 Report

The author's made changes in the Ms, improving the quality. I recommend to accept.